# Organic Electrochemical Transistor with MoS_2_ Nanosheets Modified Gate Electrode for Sensitive Glucose Sensing

**DOI:** 10.3390/s23177449

**Published:** 2023-08-27

**Authors:** Jin Hu, Jiajia Dai, Caiping Huang, Xierong Zeng, Weiwei Wei, Zhezhe Wang, Peng Lin

**Affiliations:** 1Shenzhen Key Laboratory of Special Functional Materials & Guangdong Research Center for Interfacial Engineering of Functional Materials, College of Materials Science and Engineering, Shenzhen University, Shenzhen 518060, China; 2151120226@email.szu.edu.cn (J.H.); jjd886@163.com (J.D.); 2015200119@email.szu.edu.cn (C.H.); zengxier@szu.edu.cn (X.Z.); 2Fujian Provincial Key Laboratory of Quantum Manipulation and New Energy Materials, College of Physics and Energy, Fujian Normal University, Fuzhou 350117, China; zzwang@fjnu.edu.cn

**Keywords:** organic electrochemical transistor, MoS_2_ nanosheets, liquid phase ultrasonic exfoliation, glucose sensor

## Abstract

An organic electrochemical transistor (OECT) with MoS_2_ nanosheets modified on the gate electrode was proposed for glucose sensing. MoS_2_ nanosheets, which had excellent electrocatalytic performance, a large specific surface area, and more active sites, were prepared by liquid phase ultrasonic exfoliation to modify the gate electrode of OECT, resulting in a large improvement in the sensitivity of the glucose sensor. The detection limit of the device modified with MoS_2_ nanosheets is down to 100 nM, which is 1~2 orders of magnitude better than that of the device without nanomaterial modification. This result manifests not only a sensitive and selective method for the detection of glucose based on OECT but also an extended application of MoS_2_ nanosheets for other biomolecule sensing with high sensitivity.

## 1. Introduction

Diabetes, a group of metabolic diseases fueled by the prevalence of obesity and an unhealthy lifestyle, has become a major cause of premature mortality or complication diseases including blindness, heart attacks, strokes, and numbness, according to the World Health Organization (WHO) [1,2,3]. Although there is no permanent cure for diabetes, timely glucose level monitoring and then appropriate medication have been regarded as effective methods to hinder complications and reduce the severity of the disease. So far, several types of biosensors have been developed to detect glucose, with the research direction moving from invasive to wearable [4,5,6]. Methods for glucose detection include those based on spectroscopy [7,8], colorimetry [9,10], electrochemistry [11,12,13], and so on. Electrochemical sensors, in particular, have drawn much attention because of their simple equipment, easy operation, fast response time, and high sensitivity. Yet due to working in a complex aqueous environment, the glucose sensor can be mired by humidity, pH, and other chemicals, leading to relatively poor sensitivity and selectivity. Organic electrochemical transistor (OECT), a new branch of organic thin-film transistor, seems to overcome the constraints and show a promised sensing platform due to its stability in aqueous environment operation [14,15,16,17]. It not only combines the advantages of traditional electrochemical detection but also the signal amplification function, and thus has a very high sensitivity with a low detection limit. In addition, the superiority of low working voltage, good biocompatibility, easy fabrication, and potentially flexible form all make OECT an efficient application for biomolecules (such as ions [18,19], biomarkers [20,21], cells [22,23]) and electrophysiology [24,25,26]. Therefore, OECT shows an ideal platform for highly sensitive glucose sensing, which is crucial to the diagnosis of diabetes.

Most commercialized glucose sensors are based on a glucose-enzyme reaction, with their products detected by an amperometric electrode [27]. Thus, the mechanism of sensing in OECT involves boosting the process of the enzymatic reactions. For this purpose, the sensitivity and selectivity can be improved generally by increasing the surface area of the enzymatic electrooxidation reaction or promoting the catalytic activity of the enzyme, which ultimately accelerates the electron transport to the gate electrode and thus an electric signal change in the OECT device. For example, Macaya et al. [28] fabricated an enzymatic OECT with Pt wire, which was coiled to increase the reaction area, leading to a micromolar sensitivity within the clinical range of the glucose level in human saliva. Noble metal nanoparticles (NPs), such as Pt NPs, were also modified for the enzyme electrode to improve the sensitivity of OECT for glucose, owing to NPs’ large specific area, excellent conductivity, and electrocatalytic properties [29,30]. Due to its fascinating structures (high specific surface area and nanosheet morphology) and electronic property (excellent electron mobility), two-dimensional material (2DM) has become a good candidate for electrode modification for OECT to promote the redox reaction and thereby improve the sensitivity and selectivity for glucose detection [31,32,33]. We anticipate that 2DM may be a potential strategy for modification of the OECT gate for glucose detection.

Molybdenum disulfide (MoS_2_), a metal sulfide two-dimensional nanomaterial, has attracted exuberant attention in the area of electrochemical biosensors due to its large surface area, tunable band gap, high electron mobility, excellent electrocatalytic properties, and ion-intercalation surface morphologies [34,35]. MoS_2_, with its graphene-like structure, has a high surface-to-volume ratio, and its superior layer spacing (0.65 nm) compared to graphene (0.34 nm) facilitates the probability of intercalation of a large number of small biomolecules [36]. The MoS_2_ obtained can promote redox transfer between the enzyme and the electrode surface. Moreover, there are catalytic active sites at the edge of the MoS_2_ nanometer lamellar structure [37], which can be used as an electrochemical catalyst for the chemical reactions between the enzyme and detected biomolecules, thereby improving the sensitivity and detection limit of the devices [38,39,40]. The satisfactory analytical properties evidence that the use of MoS_2_ modified on the reaction electrode holds great promise for detecting glucose of interest. However, the properties of MoS_2_ are susceptible to being affected by the preparation methods, resulting in slow progress in the application field of biological sensing. Therefore, to fully exploit the superiority of MoS_2_, the implementation of an optimal production method to improve the sensitivity and selectivity of biosensors is needed.

In this paper, we report a kind of OECT-based glucose sensor in which MoS_2_ nanosheets were modified on the gate electrode for improving the sensitivity and detection limit. MoS_2_ nanosheets were exfoliated in different solvents for ease of surface functionalization and modification of the gate electrode of OECT. Nafion was also modified to facilitate the immobilization of MoS_2_ nanosheets and an enzyme (glucose oxidase, GOx). Nafion, a sulfonated copolymer enjoying good film-forming ability and susceptibility to chemical modifications that has been widely used as a support for enzyme immobilization, could be used to noncovalently associate with the modified materials on the gate electrode, leading to a high selectivity for the glucose sensor. Compared with a non-nanomaterials-modified device, the detection limit could be increased by 1~2 orders of magnitude with high selectivity. This work manifests that the OECT sensing platform holds great promise for sensitive detection of glucose and also indicates that MoS_2_ nanosheets have great potential in biosensing applications.

## 2. Experimental Details

### 2.1. Chemicals and Apparatus

Poly(3,4-ethylenedioxythiophene):poly(styrene sulfonate) (PEDOT:PSS) aqueous solution and Nafion solution (diluted to 0.5%, 1.0%, 1.5%, and 2.0% by PBS solution) were obtained from Sigma–Aldrich Co., Shanghai, China and stored at 4 °C for future use. Molybdenum disulfide (MoS_2_, MW 160.07, 99.5%), lactic acid, and sarcosine were purchased from Aladdin Reagent Database, Inc., Shanghai, China. Phosphate buffered saline (PBS, pH = 7.4) was purchased from Thermo Fisher Scientific Co., Ltd., Shanghai, China. Glucose oxidase (GOx) (100 KU g^−1^) was acquired from J&K Scientific Ltd., Guangzhou, China and stored at –20 °C. GOx stock solution (10 mg mL^−1^) in PBS was stored at 4 °C in a refrigerator. Glucose (C_6_H_12_O_6_, AR) was purchased from Shanghai Ling-feng chemical reagents Co., Ltd., Shanghai, China. Sodium cholate (C_24_H_39_NaO_5_, MW 430.56, >98.0%), N-Methyl pyrrolidone (NMP), and ethanol were ordered from J&K Scientific Ltd., Guangzhou, China, Damao Chemical Reagent Factory, Tianjin, China, and Sinopharm Chemical Reagent Co., Ltd., Shanghai, China, respectively.

All electrochemical measurements were performed using a Keithley 2400 source meter. The morphology of MoS_2_ nanosheets was characterized by transmission electron microscopy (TEM, JEM-2100F, JEOL, Tokyo, Japan), an atomic force microscope (AFM, Bruker, Germany), and an X-ray powder diffractometer (XRD, D8 Advance, Bruker, Germany).

### 2.2. Preparation of MoS_2_ Nanosheets

The MoS_2_ nanosheets were mainly prepared by liquid-phase ultrasonic exfoliation. A certain amount of MoS_2_ powder was mixed with a solvent, milled by a ball for 12 h, and then dried at 50 °C to evaporate the solvent. In order to obtain high-quality MoS_2_ nanosheets, the NMP solution, sodium cholate solution (1.5 mg mL^−1^) and aqueous alcohol (45% VOL) were chosen as the dispersed solvents, respectively. An amount of 0.1 g of MoS_2_ powder was mixed with 20 mL of the above solvents and sonicated continuously for 8 h at 240 W to obtain a stripping solution. The stripping solution was centrifuged for 30 min at 3000 rpm, and the supernatant was decanted. The supernatant was then centrifuged again for 20 min at 12,000 rpm, and the precipitate was collected. The collected precipitate was mixed into deionized water and centrifuged again for 20 min at 12,000 rpm to remove the solvent contained in the precipitate. Finally, the clean precipitate was redistributed in deionized water for future use.

### 2.3. Preparation of OECT-Based Glucose Sensors

Figure 1 shows a schematic diagram of an OECT-based glucose sensor. First, Au/Cr film (100 nm/10 nm) was coated by the thermal evaporation method and patterned for source and drain electrodes on a glass substrate. Then the glass substrate was treated with oxygen plasma to improve the surface hydrophilicity. A PEDOT:PSS layer was spin-coated on the glass substrate, and the channel area on the glass surface was then patterned. The channel length and width of the devices were 0.2 mm and 6 mm, respectively. The OECT device was annealed at 180 °C for 1 h under N_2_.

For preparing a sensitive and selective glucose sensor, the gate electrode was modified with Nafion, GOx, and MoS_2_ nanosheets. To prepare the Nafion-GOx/Pt gate electrode, 20 μL GOx (20 mg mL^−1^) and 20 μL Nafion were mixed, and the solution was drop-coated onto the surface of Pt electrodes with a pipette and then dried at 4 °C overnight. When the Nafion film was formed, the Nafion-GOx/Pt electrodes were immersed in de-ionized water to wash away the residue and stored at 4 °C for future use. The Nafion-MoS_2_-GOx/Pt gate electrode was prepared in the same way by mixing 20 μL GOx (20 mg mL^−1^), 20 μL Nafion, and 20 μL MoS_2_ nanosheets in an aqueous solution.

### 2.4. Measurement and Characterization

The OECT devices were characterized by using two Keithley 2400 source meters. The OECT devices were placed in phosphate buffer saline (PBS, pH = 7.4), using Nafion-GOx/Pt electrodes or Nafion-MoS_2_-GOx/Pt electrodes as gate electrodes. Before testing, all prepared electrodes were immersed in PBS for 20 min to eliminate interference. The output and transfer characteristics of the device with a pure Pt gate electrode were measured in PBS solution, and the results are shown in Appendix A. Different glucose solutions were prepared and slowly added to the PBS solution for testing. In order to obtain the response of the channel current to different concentrations of glucose, the gate voltage and the channel voltage were fixed at 0.5 V and 0.1 V, respectively.

## 3. Results and Discussion

### 3.1. Characterization of MoS_2_ Nanosheets

Figure 2a,b show the XRD patterns of bulk MoS_2_ and MoS_2_ nanosheets, respectively. After ultrasonic stripping, the XRD peaks of the sample at 2θ = 15°, 45°, and 60° are attributed to the (002), (006), and (008) planes of MoS_2_, respectively. This result indicates that MoS_2_ is separated along the surface of the (00*l*) plane, and the MoS_2_ nanosheets have a good monoorientation with a multi-layer structure. TEM images of the morphology and structure of MoS_2_ nanosheets are shown in Figure 2c,d. The size of MoS_2_ nanosheets ranges from hundreds of nanometers to several micrometers, and some nanosheets have multi-layer superposition. HRTEM images further reveal that the crystal lattice structure of MoS_2_ nanosheets is hexagonal, and the d-spacing of the (100) plane is 0.262 nm. Figure 2e is an AFM image of the structural features of MoS_2_ nanosheets. It shows the same morphology as can be seen in the observations from TEM images and reveals the MoS_2_ nanosheets have a uniform thickness of 2~5 nm.

### 3.2. OECT-Based Glucose Sensor with Nafion-GOx/Pt Gate Electrode

Figure 3a shows the *I*_DS_-*T* curve of the response with a pure Pt electrode as the gate electrode of OECT to the glucose solution. As the concentration of glucose solution increased, the channel current *I*_DS_ of the OECT device did not change significantly, indicating that the pure Pt gate electrode may not be suitable for glucose detection. Therefore, GOx and Nafion were used to modify the Pt gate electrode to improve the sensitivity of the device. Figure 3b,c show the response of the OECT with a Pt gate electrode modified with GOx and Nafion (1.5%) to the additions of glucose at different concentrations. The glucose concentration was initially added at 1 nanomolar and then increased at regular intervals. It can be seen that when the concentration of glucose was 5 μM, the channel current *I*_DS_ started to show a significant step change, and the response increased significantly with the increase in glucose concentration. Compared with the device without modification of the Pt gate electrode (Figure 3a), the sensitivity of glucose detection is significantly improved. As with our previous research, the reason is the specific catalytic action of GOx on glucose [32]. When glucose is added to the electrolyte, it is catalyzed by GOx to produce H_2_O_2_. Meanwhile, H_2_O_2_ is further oxidized on the surface of the Pt gate electrode to make the electron transfer (Faraday current). And then the effective gate voltage applied to the OECT device is changed, thereby changing the channel current *I*_DS_ [32]. Hence, the enzyme GOx plays an important role in the sensitivity and selectivity of the OECT sensor. The chemical reactions on the OECT gate electrode are as follows [32,41]:(1)$$D\text{-}\mathrm{glucose}\overset{\;\mathrm{Enzyme}\left(\mathrm{GOx}\right)}{\to }D\text{-}\mathrm{glucono}\text{-}1,5\text{-}\mathrm{lactone}+H_{2}O_{2}$$
(2)$$2H_{2}O_{2}\overset{\;-2e^{-}}{\to }O_{2}+2H_{2}O$$

It can be seen that the enzyme GOx makes the sensor both sensitive and selective for the biocatalytic reaction of glucose. During the measuring period, glucose is detected by monitoring the change in channel current *I*_DS_. *I*_DS_ of the OECT is given by [15]:(3)$$I_{\mathrm{DS}}=\frac{G}{V_{p}}(V_{p}-V_{G}^{\mathrm{eff}}+\frac{V_{\mathrm{DS}}}{2})V_{\mathrm{DS}}\;\;\;(\mathrm{when}\;\left|V_{\mathrm{DS}}\right|\ll \left|V_{p}-V_{G}^{\mathrm{eff}}\right|)$$
where *G* is the conductance of the organic semiconductor film, $V_{p}$ is the pinch-off voltage, and $V_{G}^{\mathrm{eff}}$ is the effective gate voltage. Different glucose concentrations can induce corresponding potential drops, which cause a change in the effective gate voltage. The effective gate voltage can be calculated from the transfer curve of the OECT device. Figure 3b shows the relationship between effective gate voltage and glucose concentration. When the concentration of the glucose solution increases by an order of magnitude, the effective gate voltage changes by 39.82 mV.

### 3.3. OECT-Based Glucose Sensor with Nafion-MoS_2_-GOx/Pt Gate Electrode

To obtain a highly sensitive OECT glucose sensor, nanomaterials are usually modified on the gate electrode [33]. MoS_2_ nanosheets have excellent electrocatalytic performance and a large specific surface area, which can further improve the sensitivity of the device. In addition, unlike NPs and their colloidal dispersions suffering from aggregation or the high production cost of graphene, MoS_2_ nanosheets can be synthesized by liquid-phase ultrasonic exfoliation methods, which are cheaper and easier. Therefore, we explored the effects of different exfoliated solvents and sizes of MoS_2_ nanosheets on the detection performance of OECT glucose sensors. Table 1 shows the detection limit of the devices with Nafion-GOx/Pt and Nafion-MoS_2_-GOx/Pt gate electrodes, where the concentration of the modified Nafion solution on all gate electrodes is fixed at 1.5%. As can be seen, when MoS_2_ nanosheets exfoliated by aqueous alcohol were modified on the gate electrode, the device had the best detection performance with a detection limit of 0.25 μM. And the change in effective gate voltage was about 53.37 mV when the glucose concentration increased by an order of magnitude. Figure 4 shows the current response of the devices with MoS_2_ nanosheets stripped with different solvents to different concentrations of glucose. As shown in Figure 4c, there is a significant current response to 0.25 μM glucose. Compared with the device without MoS_2_ nanosheet modification, the detection limit is optimized from 5 μM to 0.25 μM, and the change of effective gate voltage is increased from 39.82 mV to 53.37 mV per decade, indicating that the sensitivity of the device with MoS_2_ nanosheet modified gate electrode is significantly improved.

In order to analyze the effect of MoS_2_ stripping solvents on detection performance, we characterized the morphology and size of MoS_2_ nanosheets prepared with different stripping solvents. As shown in Appendix A, the size of MoS_2_ nanosheets exfoliated by aqueous alcohol is smaller than that of samples prepared by other stripping solvents. It could be deduced that MoS_2_ nanosheets exfoliated by aqueous alcohol may have a larger specific surface area, which can increase the effective enzyme load on the gate electrode. In addition, the higher surface area of the thin MoS_2_ nanosheets would result in more active edge sites, which are crucial for excellent catalytic activity [37]. Overall, MoS_2_ nanosheets exfoliated by aqueous alcohol have a smaller size and thus more active sites exposed at the edges, which is conducive to increasing the catalytic reaction speed of glucose oxidase and thus improving the sensitivity of glucose detection. Therefore, for liquid-phase exfoliated methods, solvent selection plays an important role in the quality of MoS_2_ nanosheets.

In addition, the concentration of Nafion solution is also very important to the sensor’s performance. To optimize this condition, four different concentrations of Nafion solution (0.5%, 1%, 1.5%, and 2%) were used to modify the gate electrode, as shown in Table 2. Figure 5 shows the channel current response of the device modified with different concentrations of Nafion on the gate electrode. It can be seen that the OECT with Nafion (1%)-MoS_2_-GOx/Pt gate electrode reveals the best sensing performance, with glucose detection limits down to 100 nM, which are comparable to or even better than those of some other methods (LOD at the nM level). With a thinner or thicker Nafion layer on the electrode, the sensor demonstrates inferior performance in sensitivity. On one hand, the relatively thin Nafion film (0.5%) could have a poor effect on the immobilization of MoS_2_ nanosheets and the enzyme GOx, which may cause MoS_2_ nanosheets and GOx to fall off. On the other hand, a too-thick Nafion film (2%) would hinder the charge transfer to the electrode, thus leading to a weak redox reaction. Therefore, for the modification of the gate electrode in OECT, the optimal concentration of Nafion was 1%.

We consider that the high sensitivity of the device with a Nafion-MoS_2_-GOx/Pt gate electrode can be attributed to the enhancement of reaction surface area and electrocatalysis and the fast electron transfer to the gate electrode. Firstly, a certain amount of Nafion-MoS_2_ hybrid was used for GOx enzyme immobilization, in which a small amount of Nafion-MoS_2_ hybrid may not result in sufficient immobilization and too much hybrid may lead to a thick film and prohibit glucose from reaching GOx. Then, MoS_2_ nanosheets have a large surface area for enzyme immobilization and fast mass transport due to the large interlayer distance of their ordered structure. The open structures of the Nafion-MoS_2_-GOx enzyme electrode made the enzyme electrode highly catalytic, strengthening the electrochemical reaction between GOx and glucose. At last, owing to the electrochemically catalytic activity of MoS_2_ nanosheets, the realization of fast electron transfer to the gate electrode is of great significance for sensitivity.

### 3.4. Selectivity of OECT-Based Glucose Sensor

In fact, there are many interfering biomolecules in blood, urine, saliva, and other body fluids for practical glucose detection, which would result in a false diagnosis of glucose content. Therefore, it is necessary to study the selectivity of OECT-based glucose sensors to meet the requirements of practical application. Some commonly coexisting interfering substances, such as lactic acid and sarcosine, in human biological fluids are examined [42,43]. Interference experiments were conducted with PBS solution, lactic acid, sarcosine, and glucose added in the reaction process in order, and the concentrations of all analytes were fixed at 10 mM. A sufficient amount of analyte was applied for better signal response so as to verify the selectivity. Figure 6 shows the selectivity of the device with a Nafion (1%)-MoS_2_-GOx/Pt gate electrode. As can be seen, in the absence of glucose, the other three interfering biomolecules led to negligible changes, and only the glucose solution was essential for a substantial change in channel current *I*_DS_. The result indicates that the proposed sensing platform shows a high degree of selectivity toward glucose. Theoretically, the enzyme GOx only reacts electrochemically with glucose, leading to a change in the current signal of OECT. As for the biomolecules catalyzed directly by the Pt gate electrode, such as dopamine, the common practice is to modify some organic membranes with electrostatic repulsion effects to improve the selectivity of the sensor, which is beyond the scope of this work.

## 4. Conclusions

In summary, OECTs with enzyme (GOx), Nafion (1%), and MoS_2_ nanosheets modified on the Pt gate electrodes demonstrate good performance in glucose sensing. MoS_2_ nanosheets were prepared by liquid-phase ultrasonic exfoliation with the optimum exfoliation solvent, and both the size and amount of MoS_2_ nanosheets were obtained based on the results of glucose detection. Nafion was also modified to facilitate the immobilization of MoS_2_ nanosheets and the enzyme GOx, which promoted MoS_2_ functionalization and thus improved the selectivity of the sensor. The device with a Nafion-MoS_2_-GOx/Pt gate electrode showed a low detection limit of 100 nM and a high selectivity for glucose, with a change in effective gate voltage of 50.37 mV/decade. Summarily, this work not only provides a high sensitivity and selectivity method for the detection of glucose but also extends the modification application of MoS_2_ for the detection of other biomolecules.

## Figures and Tables

**Figure 1 sensors-23-07449-f001:**
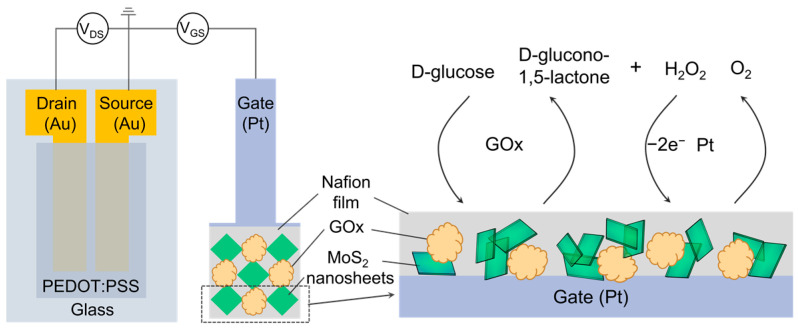
Schematic diagram of the MoS_2_ modified OECT-based glucose sensor.

**Figure 2 sensors-23-07449-f002:**
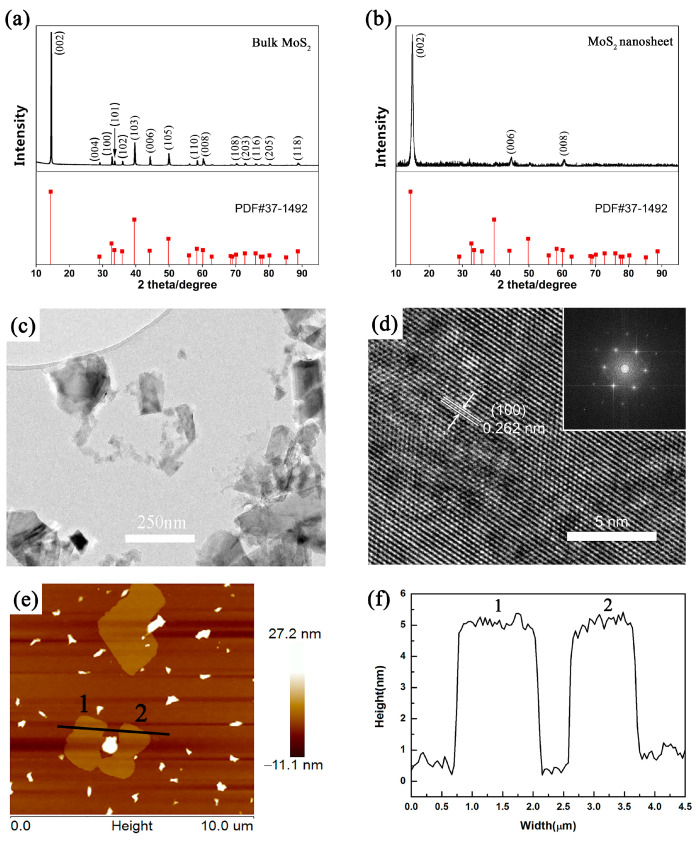
XRD images of (**a**) MoS_2_ powder and (**b**) MoS_2_ nanosheets. (**c**) TEM image of MoS_2_ nanosheets stripped in aqueous alcohol. (**d**) HRTEM image of MoS_2_ nanosheets stripped in aqueous alcohol. (**e**) AFM image of MoS_2_ nanosheets stripped in aqueous alcohol. (**f**) Thickness of positions 1 and 2 in AFM.

**Figure 3 sensors-23-07449-f003:**
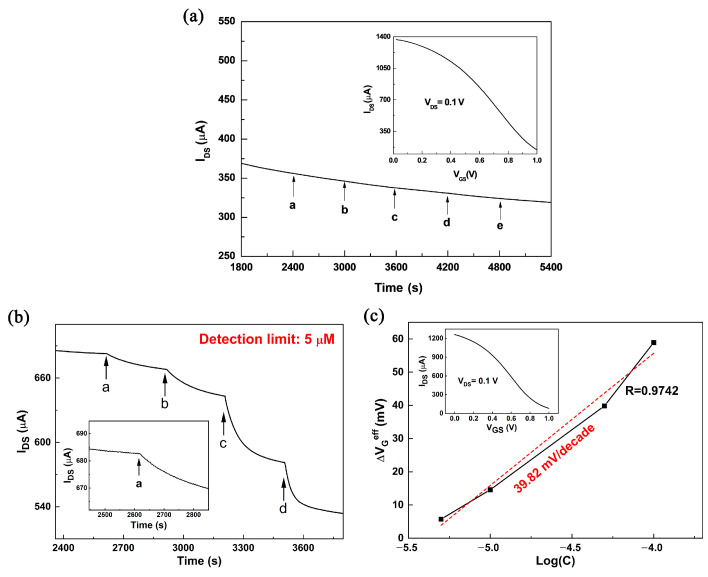
Channel current response of OECT with (**a**) a pure Pt gate electrode, (**b**) Nafion (1.5%)-GOx/Pt gate electrode, for glucose detection in PBS solution. *V*_G_ is fixed at 0.5 V. The concentrations of glucose added: in (**a**), from a to e: 0.5, 1, 5, 10 and 100 μM; in (**b**), from a to d: 5, 10, 50 and 100 μM. Inset: transfer curve (*I*_DS_ vs. *V*_G_) of the device characterized at *V*_DS_ = 0.1 V. (**c**) The change of effective gate voltage (${∆V}_{G}^{\mathrm{eff}}$) in (**b**) versus the concentrations of glucose. Inset: transfer curve (*I*_DS_ vs. *V*_G_) of the device characterized in PBS solution.

**Figure 4 sensors-23-07449-f004:**
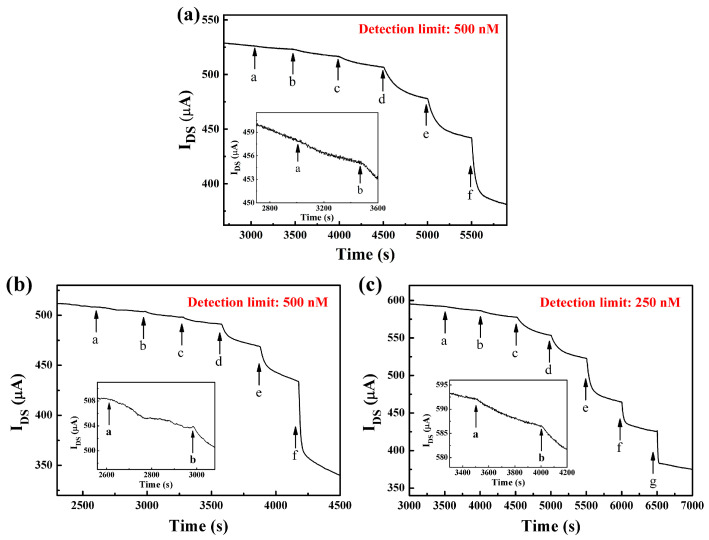
Channel current response of an OECT with Nafion (1.5%)-MoS_2_-GOx/Pt gate electrodes for glucose detection in PBS solution. The modified MoS_2_ nanosheets are exfoliated by (**a**) NMP solution, (**b**) sodium cholate solution, and (**c**) aqueous alcohol. The concentrations of glucose added: in (**a**), from a to f: 0.5, 1, 5, 10, 50, and 100 µM; in (**b**), from a to f: 0.5, 1, 5, 10, 50, and 100 µM; in (**c**), from a to g: 0.25, 0.5, 1, 5, 10, 50, and 100 µM.

**Figure 5 sensors-23-07449-f005:**
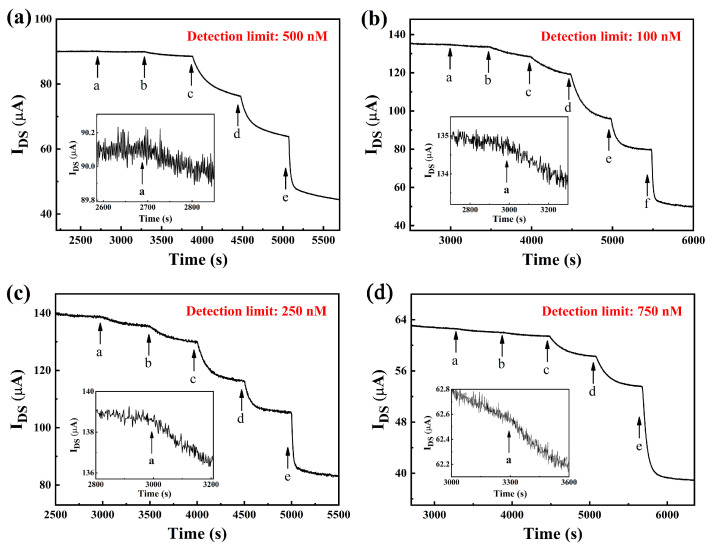
Channel current response of OECT with Nafion-MoS_2_-GOx/Pt gate electrodes for glucose detection in PBS solution. The Nafion solution concentrations of the different Nafion-MoS_2_-GOx films on the Pt gate electrode are (**a**) 0.5%, (**b**) 1%, (**c**) 1.5%, and (**d**) 2%. The concentrations of glucose added: in (**a**), from a to e: 0.5, 1, 5, 10 and 100 µM; in (**b**), from a to f: 0.1, 0.5, 1, 5, 10 and 100 µM; in (**c**), from a to e: 0.25, 1, 5, 10 and 100 µM; in (**d**), from a to e: 0.75, 1, 5, 10 and 100 µM.

**Figure 6 sensors-23-07449-f006:**
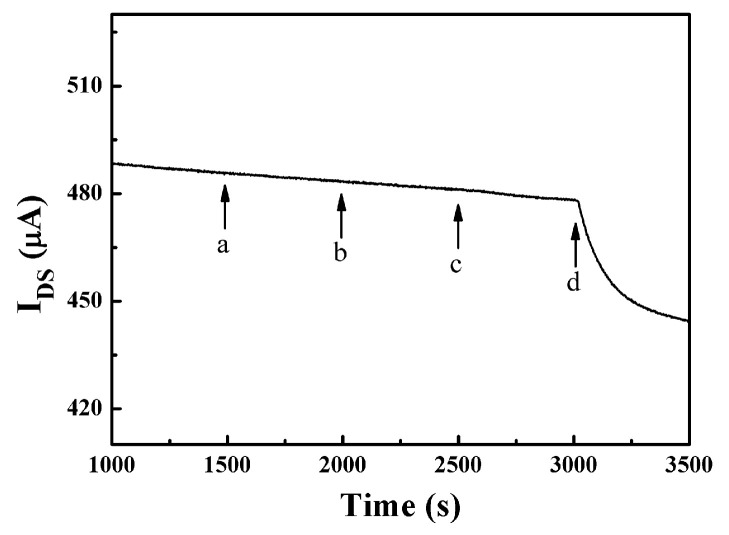
Selectivity of an OECT with Nafion (1%)-MoS_2_-GOx/Pt gate electrode. *V*_G_ is fixed at 0.5 V. From a–d, the added analytes are PBS solution, lactic acid, sarcosine, and glucose, respectively. The concentrations of all analytes are 10 mM.

**Table 1 sensors-23-07449-t001:** Detection limit and the change of effective gate voltage (α) of the devices with MoS_2_ nanosheets stripped with different solvents and 1.5% Nafion membrane-modified gate electrodes.

Gate Electrode	Stripping Solvent	Detection Limit (µM)	α (mV/Decade)
Nafion (1.5%)-GOx/Pt	Nil	5	39.82
Nafion (1.5%)-MoS_2_-GOx/Pt	NMP solution	0.5	47.58
Nafion (1.5%)-MoS_2_-GOx/Pt	Sodium cholate solution	0.5	48.09
Nafion (1.5%)-MoS_2_-GOx/Pt	Aqueous alcohol	0.25	53.37

**Table 2 sensors-23-07449-t002:** Detection limit and the change of effective gate voltage (α) of the OECT-based sensors for glucose detection.

Gate Electrode	Detection Limit (nM)	α (mV/Decade)
Nafion (0.5%)-MoS_2_-GOx/Pt	500	43.20
Nafion (1%)-MoS_2_-GOx/Pt	100	50.37
Nafion (1.5%)-MoS_2_-GOx/Pt	250	45.92
Nafion (2%)-MoS_2_-GOx/Pt	750	33.80

## Data Availability

Not applicable.

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
