# Peer review of "Organic Electrochemical Transistor with MoS2 Nanosheets Modified Gate Electrode for Sensitive Glucose Sensing"

_sensors, 2023, doi:10.3390/s23177449_

Round 1

Reviewer 2 Report

The paper describes an organic electrochemical transistor (OECT) with MoS2 nanosheets modified on the gate electrode was proposed for glucose sensing. MoS2 nanosheets show excellent electrocatalytic  performance, large specific surface area and more active sites and were prepared by newly modified method of sonic exfoliation to modify the gate electrode of OECT, resulting in a large improvement in the sensitivity of the glucose sensor.

General recommendations:

1. In the introduction please explain what is Nafion and why do you use in the present work.

2. It is worth mentioning that the glucose concentration in human blood is in the mM range and such 500 nm detection limits are not needed. I do not know the concentrations in saliva and perspiration where the real applications of this proposal may be.

3. In order to obtain electrochemical signal from GOx to glucose 3 conditions have to be met: a)glucose has to reach the GOx; b) the GOx must have enough space around it (not be constrained) in order for the reaction to take place; and c) the obtained current should be able to reach the electrode. A discussion of all these and why it is achieved in the nanosheet modified sensing layer would improve the quality of the paper. Also, I highly recommend on adding a drawing on how the surface layer looks like. 

Specific comments for the corresponding lines:

Line 168 - This should be Fig. 2, not 1.

Line 70-71 The size of GOx is around 5 nm. How increasing the spacing in the subnanometer range will help immobilize more GOx significantly?

79 - The presence of defects can actually increase sensor sensitivity due to increased surface area. Please bring another motivation for this research.

 85 - Nafion is first mentioned here. Please bring context - why you use it, what it is. I had to look the Wikipedia to get the context!

130 - why annealed?

135 - coated - how? Dip-coated, spray-coated, LB film deposition, etc.

184 - I would suggest moving Fig. S2 into the main article and comparing the 2 graphs on a single figure.

Fig. 6 - Why there is drift in the current over time.
